# Ethyl Gallate Isolated from *Castanopsis cuspidata* var. *sieboldii* Branches Inhibits Melanogenesis and Promotes Autophagy in B16F10 Cells

**DOI:** 10.3390/antiox12020269

**Published:** 2023-01-25

**Authors:** Moon-Hee Choi, Seung-Hwa Yang, Da-Song Kim, Nam-Doo Kim, Hyun-Jae Shin

**Affiliations:** 1Department of Beauty and Cosmetology, Graduate School of Industrial Technology and Entrepreneurship, Chosun University, Gwangju 61452, Republic of Korea; 2Department of Chemical Engineering, Graduate School of Chosun University, Gwangju 61452, Republic of Korea; 3VORONOI BIO Inc., Incheon 21984, Republic of Korea

**Keywords:** *Castanopsis cuspidata* var. *sieboldii*, ethyl gallate, antimelanogenesis, AKT pathway, autophagy enhancer

## Abstract

The *Castanopsis cuspidata* var. *sieboldii* (CCS) plant grows predominantly in temperate regions of Asian countries, such as South Korea. Research on CCS has so far concentrated on the nutritional analysis, antioxidant activity, and anti-inflammation properties of its branches. However, the isolation of compounds and structural elucidation of effective single molecules remain unexplored, necessitating further exploration of CCS branches. Therefore, this study demonstrates the antioxidant and antimelanogenic activity of a single substance of ethyl gallate (EG) isolated from CCS branch extracts. Notably, the antimelanogenic (whitening) activity of EG extracted from CCS branches remains unexplored. Tyrosinase inhibition, kinetic enzyme assays, and molecular docking studies were conducted using mushroom tyrosinase in order to examine the antioxidant mechanism and antimelanin activity of EG in B16F10 melanoma cells. Nontoxic EG concentrations were found to be below 5 µg/mL. While EG significantly reduced the levels of whitening-associated proteins, p-CREB, and p-PKA, it dose-dependently inhibited the expression of TYR, TRP-1, TRP-2, and transcription factor (MITF). In addition, EG downregulated melanogenetic gene expression and activated autophagy signals. Therefore, EG extracted from CCS branches could serve as a novel functional cosmetic material with antimelanogenic and autophagy-enhancing activity.

## 1. Introduction

As interest in well-being, health, and life expectancy grows, so does the value of healthy aging and clean skin. Various plant-derived natural compounds are currently used as functional components in cosmetics. Among the various functions of cosmetics, antioxidant activity is a significant factor in the skin’s UV defense mechanism. Notably, the skin’s inherent exposure to UV radiation and air induces the generation of radical and nonradical reactive oxygen species (ROS) via reactions with molecules [1]. ROS accelerates skin aging by destroying skin antioxidants and initiating lipid peroxidation, protein oxidation, DNA oxidation, chain scission, and abnormal cross-linking of collagen and hyaluronic acid, resulting in wrinkles and melanin production [2,3,4]. Because excessive UV radiation can overwhelm the skin defense system, appropriate antioxidants capable of suppressing ROS hyperproduction are needed [5,6,7,8]. Research on diverse natural products is increasing in an effort to produce effective natural antioxidants as alternatives to synthetic ones [9]. In addition to the discovery of cosmetic ingredients with antiaging and whitening activity, a novel autophagy modulator (enhancer or inhibitor) has also been developed [10]. Autophagy is a novel antiaging phenomenon essential for cells to remove intracellular waste products or unnecessary proteins and recycle them to regenerate cells in a healthy state, implying that uncovering the role of autophagy in the whitening mechanism could provide a new breakthrough in melanogenesis research.

*Castanopsis cuspidata* var. *sieboldii* (CCS), a typical South Korean native plant, possesses excellent carbon storage and absorption capacity. It is also distributed in Japan, China, and Taiwan. Previously reported components of CCS include galloylshimic acid [11], hydrolyzable tannins, terpenoids [12], castanopsinin, ellagitannin [13], dehydrodialic acid, creatinine, chestanine [14], galloyl ester triterpenoid, and hexahydroxydiphenic acid [15]. Notably, prior to this study, neither the isolation of ethyl gallate (EG) nor elucidation of whitening mechanisms related to autophagy signals had been described. Therefore, this study identified novel compounds and investigated the whitening activity, autophagy enhancement, and intracellular signal transduction mechanism of CCS branch extracts and solvent fractions in order to ascertain CCS potential as a whitening cosmetic material.

## 2. Materials and Methods

### 2.1. Chemicals

*n*-Hexane, chloroform, ethyl acetate (EtOAc), n-butanol, ethanol, and methanol were obtained from OCI (SEL, KR). Folin–Ciocalteu reagent (for total phenolics), 2,2-diphenyl-1-picrylhydrazyl (DPPH), 2,2′-Azino-bis (3-ethylbenzthiazoline-6-sulfonic acid) (ABTS), dimethyl sulfoxide-d6 (DMSO, exclusive solvent for NMR), mushroom tyrosinase (EC 1.14.18.1), and ascorbic acid were obtained from Sigma Aldrich (St. Louis, MO, USA). All reagents used were of analytical grade. Phospho-CREB (p-CREB), phospho-PKA (p-PKA), AMPK, p-AKT, AKT, p-mTOR, mTOR, Becline, and LC3B were purchased from Cell Signaling (Danvers, MA, USA). Antibodies against tyrosinase (TYR), TRP-1, TRP-2, microphthalmia-associated transcription factor (MITF), CREB, PKA, α-melanocyte-stimulating hormone (α-MSH), and β-actin were purchased from Santa Cruz Biotechnology (Dallas, TX, USA). Horseradish-peroxidase-conjugated anti-mouse, anti-goat, and anti-rabbit antibodies were purchased from Invitrogen (Carlsbad, CA, USA).

### 2.2. Preparation of CCS Extracts and Solvent Fractions

In April 2021, CCS branches were collected from Wando Arboretum (Wando, Korea) in Jeollanam-do, washed, dried with hot air (40 °C), stored, and then pulverized for use. To confirm the antioxidant activity of each solvent fraction of the pine CCS’s branches, 1.5 kg of dried at 40 °C and pulverized CCS’s branches was added to 15.0 L of 70% EtOH (*v*/*v*) and immersed for two weeks at room temperature for extraction. The immersed sample was filtered using a vacuum pump and Whatman No. 1 filter. After filtration, it was separated and repeated once under the same conditions. The filtrate obtained by filtration was concentrated using a rotary vacuum evaporator in a water bath at 37 °C and then lyophilized for use. In addition, 269.03 g of 70% EtOH extract was suspended in 500 mL distilled water and fractionated sequentially in order of polarity using a separation funnel to obtain n-hexane (n-Hex), chloroform (Ch), ethyl acetate (EtOAc), n-butanol (n-BuOH), and water (H_2_O) fractions of 13.03, 5.27, 51.22, 44.74, and 15.93 g, respectively (Figure 1). Fractionation was repeated three times using 1 L of each solvent. Each fraction was used after being concentrated and lyophilized as described above, and the extraction yield was calculated using the following formula.
Yield (%) = (weight of sample after freeze drying (g)/weight of sample before extraction (g)) × 100

### 2.3. Separation of Active Components from EtOAc Fractions of CCS Branches

Using medium-pressure liquid chromatography (MPLC, Yamazen, Osaka, Japan), the EtOAc fraction was sequentially subdivided according to polarity among the solvent fractionation layers. After dissolving 38.22 g of the EtOAc fraction in 20 mL of MeOH and filtering the solution with a syringe filter, the separation experiment was conducted using an ODS column at a flow rate of 10 mL/min and an absorbance value of 280 nm. The solvent conditions included fractionation by sequentially increasing the MeOH ratio by 5% at a solvent ratio of H_2_O: MeOH (10–100% MeOH, 570 min) and MeOH (100%, 20 min) under isocratic conditions. Each fraction pattern was confirmed by thin-layer chromatography, and then the fractions were divided into three fractions (Fr.1–Fr.3) based on their antioxidant activity and subjected to HPLC and MPLC for qualitative analysis and continuous fractionation under gradient conditions, respectively (Appendix A). The MPLC fraction (Fr.2) with the highest antioxidant activity was subjected to MPLC under the solvent conditions of H_2_O: MeOH (10–100% MeOH, 80 min) and MeOH (100%, 20 min), and Prep-Compound 1 (47 mg) was then isolated by LC. Compound 1 was isolated by preparative high-performance liquid chromatography (Prep 150 LC, Waters, MA, USA) with an X-Bridge Prep OBD C18 column (5.0 µm, 19 mm × 150 mm). Elution was performed with a linear gradient of methanol (0 min, 50/50; 30 min, 100/0; 100 min, 100/0). AVANCE III HD 400 (FT-NMR system, 400 MHz, BRUKER) was used as the nuclear magnetic resonance (NMR) spectrometer for the structural analysis of the active component. Samples were prepared and analyzed at a concentration of 10 mg/mL using dimethyl sulfoxide-d6 (DMSO-d_6_) (Sigma Aldrich’s exclusive NMR solvent, St. Louis, MO, USA) as a solvent for NMR measurement.

### 2.4. Antioxidant Activity Assay

#### 2.4.1. Measurement of DPPH Free Radical Scavenging Activity

With a few modifications, Blois’ method was used to test the DPPH free radical scavenging activity [16]. For each concentration, 200 μL of sample was placed in an Eppendorf (EP) tube, and 800 μL of 0.5 mM DPPH in MeOH (1,1-diphenyl-2-picrylhydrazyl; Sigma-Aldrich Co., St. Louis, MO, USA) reagent was mixed by vortexing for 15 min. Absorbance was then measured at 517 nm using a Biotek Synergy HT multidetection microplate reader. Each sample was repeated three times to obtain an average value, and ascorbic acid and EG were used as positive controls for comparative experiments. The radical scavenging ability of each solution was calculated using the following formula and expressed as a percentage:Radical scavenging activity (%) = (Abs_control_ − Abs_sample_)/Abs_control_ × 100

#### 2.4.2. Measurement of ABTS Cation Radical Scavenging Activity

ABTS cation radical scavenging ability was measured using Liang et al.’s method, with slight modifications [17]. The reaction of an equal volume of 7 mM ABTS (2,2′-azino-bis (3-ethylbenzothiazoline-6-sulfonic acid) diammonium salt) solution with 2.45 mM potassium persulfate in the dark for 18 h produced an ABTS cationic radical. The prepared solution was diluted with distilled water until the absorbance value at 730 nm reached 0.90 ± 0.02. For each concentration, 200 μL of the sample was placed in an EP tube, mixed with 1000 μL of a 10-fold diluted ABTS solution, and vortexed in a dark room. Following a 15 min reaction, absorbance was measured at 730 nm using a Biotek Synergy HT multidetection microplate reader; each sample was repeated 3 times to obtain an average value, and ascorbic acid and EG were used as positive controls for comparative experiments. The radical scavenging activity of each solution was calculated using the following formula and expressed as a percentage, and the sample concentration (IC_50_) was obtained when the radical scavenging activity percentage of each sample was 50%.
Radical scavenging activity (%) = (Abs_control_ − Abs_sample_)/Abs_control_ × 100

### 2.5. Total Polyphenol Content Analysis

Total polyphenol content analysis was performed using the Folin–Ciocalteu method [18]. Gallic acid was employed as a reference material in a standard calibration curve for quantitative analysis, and the R^2^ value of this curve was at least 0.99. After taking 500 μL each of a sample of 1 mg/mL concentration and diluted gallic acid standard solution for each concentration, 500 μL of 0.2 M Folin–Ciocalteu’s phenol reagent and 500 μL of 2% sodium carbonate aqueous solution (*w*/*v*) were also mixed in a dark room. Following a 30 min reaction, absorbance was measured at 750 nm using a Biotek Synergy HT multidetection microplate reader. The measured value was converted into the amount of gallic acid (GAE) contained per 1 g of the sample by substituting it into a standard calibration curve to obtain the total polyphenol content.

### 2.6. Analysis of Total Flavonoid Content

Total flavonoid content was measured using Park et al.’s method, with slight modifications [19]. Quercetin was employed as a reference material in a standard calibration curve for quantitative analysis, and the R^2^ value of this curve was at least 0.99. After taking 500 μL each of a sample of 1 mg/mL concentration and diluted quercetin standard solution for each concentration, 1.5 mL of methanol, 100 μL of 10% aluminum chloride, 100 μL of 1 M potassium acetate, and 2.8 mL of distilled water were added in that order to 40 μL at room temperature and reacted for min. Absorbance was then measured at 415 nm using a Biotek Synergy HT multidetection microplate reader. The measured value was converted into the amount of quercetin (QUE) contained per 1 g of the sample by substituting it into a standard calibration curve to obtain the total flavonoid content.

### 2.7. Quantitative Analysis of Polyphenols Using HPLC MS/MS

The LC-MS/MS analysis instrument used in this experiment was an AB SCIEX 4000 Q Trap LC/MS/MS System (Shimadzu LC 20A System, Kyoto, Japan), and water (in 0.1% formic acid, solvent A) was used as the mobile phase in the analysis conditions, whereas acetonitrile (in 0.1% formic acid, solvent B) was used under isocratic conditions (35% B) (Appendix A). Using Turbo Ion Spray, the analytical conditions of MS/MS were examined in both negative and positive modes.

### 2.8. Antimelanogenic Activity Assay

#### 2.8.1. Tyrosinase Inhibition Assay

The ability to inhibit tyrosinase activity was measured spectroscopically by partially modifying Choi et al.’s method (2018) [20]. Briefly, 400 μL of 0.1 mM sodium phosphate buffer (pH 6.8) was used as the buffer, whereas 200 μL of 1.5 mM L-tyrosine was used as the substrate. After adding 200 μL of each sample (CCSB-Hex, CCSB-CL, CCSB-EtOAc, CCSB-BuOH), 100 μL of tyrosinase at a concentration of 1750 units was added to prepare a mixture. Ascorbic acid and EG were used as positive controls, and after reacting 900 μL of the mixture at 37 °C for 30 min, the absorbance at 475 nm was measured using a microplate reader (Infinite M200, Tecan, San Jose, CA, USA). The ability to suppress tyrosinase activity was determined using the following formula:Inhibition rate (%) = (1−absorbance of sample added group)/absorbance of non-added group × 100

#### 2.8.2. Enzyme Kinetics Assay

In this study, Fan et al.’s enzyme kinetic experiment was modified and continued using L-DOPA as the substrate, with experiments conducted at concentrations of 0.5, 1.0, 1.5, and 2.0 mM [21]. Tyrosinase inhibition activity was detected using a spectrophotometer (Synergy HT, BIO-TEX, Winooski, VT, USA). The 96-well microplates were loaded with 80 µL aliquots of a solution containing 50–200 U/mL mushroom tyrosinase (Sigma Aldrich, St. Louis, MO, USA). Then, 80 µL of the substrate and 80 µL of EG (0.2–1.0 mM) were added. The absorbance of the 96 wells was measured at 510 nm (T0) using a microplate reader (Synergy HT, BIO-TEX, Winooski, VT, USA). Subsequently, the microplates were incubated at 25 ± 1 °C for 15 min, and the absorbance was measured again (T1). An additional reaction period of 15 min at 25 ± 1 °C was allowed, after which a new spectrophotometric reading was completed (T2). The inhibitory percentages at the two time points (T1 and T2) were obtained based on the following formula:IA% = (c − S)/c × 100,
where IA% is the inhibitory activity, C is the negative control absorbance, and S is the sample or positive control absorbance (the absorbance at time T0 subtracted from the absorbance at time T1 or T2) [22].

### 2.9. Molecular Docking Procedure

To predict the binding sites of human tyrosinase to EG, molecular docking was performed using the Glide module of the Schrodinger Package [22,23]. The X-ray crystal structure of tyrosinase (PDB ID: 2Y9X) was retrieved from the Protein Data Bank (http://www.rcsb.org, accessed on 10 October 2022). The retrieved protein structures were processed using Protein Preparation Wizard in the Schrodinger package to remove the crystallographic water molecules, add hydrogen atoms, and assign protonated states and partial charges. The missing side chains and loops were built and refined using the Prime tool of the Schrodinger suite [24]. All protein residues were parameterized using the OPLS3e force field [25,26]. Finally, restrained minimization was performed until the converged average root mean square deviation of heavy atoms was 0.3 Å.

### 2.10. Cell Culture

The B16F10 melanoma cells used in this experiment were purchased from the American Type Culture Collection (Rockville, MD, USA) [27]. The medium used for cell culture was maintained in Dulbecco’s modified Eagle’s medium (HyClone, MA, USA) supplemented with 10% fetal bovine serum (HyClone, MA, USA). After adding 50 units/mL penicillin to the medium, the experiment was conducted in a 37 °C plus 5% CO_2_ environment.

### 2.11. MTT Cell Viability Assay

Cell viability measurement experiments were conducted according to Carmichael’s method [28]. After distributing 0.18 mL of melanoma cells (B16F10) to test toxicity on a 96-well plate at a density of 1 × 10^5^ cells/well, add 0.02 mL of sample solution prepared by concentration and incubate for 48 h at 37 °C in a 5% CO_2_ incubator. The control group was cultured under the same conditions by adding the same amount of serum-free medium as the sample solution. A total of 300 µL of MTT solution prepared at 2.5 mg/mL was added to this and incubated for 1 h in a 37 °C, 5% CO_2_ incubator; the culture medium was then removed; 100 µL of DMSO was added to each well and reacted at room temperature for 30 min; and microplate absorbance at 540 nm was measured with a reader. Cell viability was expressed as the rate of decrease in absorbance of the sample-added and non-added groups.
$$\mathrm{cell}\;\mathrm{viability}\left(\%\right)=\left(1-\frac{\mathrm{Absorbance}\;\mathrm{of}\;\mathrm{sample}\;\mathrm{added}\;\mathrm{group}}{\mathrm{Absorbance}\;\mathrm{of}\;\mathrm{the}\;\mathrm{non}-\mathrm{additive}\;\mathrm{group}}\right)\times 100$$

### 2.12. Measurement of Melanin Content

To measure the melanin content for EG, Hosoi et al.’s method was modified and implemented [29]. B16F10 cells were cultured at 1 × 10^4^ cells/cm in 6-well plates. After 24 h, the cells were stimulated with α-MSH 1 µg/mL. Simultaneously, various concentrations of EG (1–5 μg/mL) were added for 48 h. After washing with phosphate-buffered saline (PBS), the cells were harvested by trypsin treatment. The collected cells were dissolved in 100 µL of 1 N NaOH and measured at 405 nm using a spectrophotometer.

### 2.13. Measurement of ROS Production

2′,7′-dichlorofluorescein diacetate (DCFH-DA) fluorescent probe was used to measure the production of intracellular ROS. Cells were pretreated with EG (1 or 5 µg/mL) for 24 h, followed by staining with 10 µM DCFH-DA for 30 min at 37 °C in the dark. The cells were subsequently washed with PBS and scraped from the well. Productions of intracellular ROS were observed under a fluorescence microplate reader (Gemini, Molecular Devices, Sunnyvale, CA, USA) at excitation/emission wavelengths of 485 nm/530 nm.

### 2.14. Immunoblot Analysis

Protein extraction, SDS-polyacrylamide gel electrophoresis, and immunoblot analysis were performed as previously reported [26]. Briefly, cell lysates were separated by SDS-PAGE (7.5%, 12% acrylamide gels) and electrophoretically transferred to a nitrocellulose (NC) membrane (GE Healthcare, Ord, IL, USA). Subsequently, the NC membranes were blocked with 5% skim milk at 37 °C and incubated overnight at 4 °C with the primary antibody. After removing the primary antibody, the NC membranes were washed three times with PBS for 10 min, followed by incubation with a secondary antibody (Invitrogen) for 1 h at room temperature. After washing, the membranes were treated with an enhanced chemiluminescence (ECL) detection kit (Amersham Biosciences, Buckinghamshire, UK). Immunoreactive protein expression was visualized using LAS 4000 (Fujifilm, Tokyo, Japan); β-actin was used as an immunoblotting control.

### 2.15. RNA Isolation and RT-PCR

TRIzol (Invitrogen) was used to obtain the total RNA extract according to the manufacturer’s protocol. To synthesize cDNA, total RNA (2 µg) was reverse-transcribed using oligo dT18 primer. The synthesized cDNA was amplified using a high-capacity cDNA synthesis kit (Bioneer, Daejeon, Korea) with a thermal cycler (Bio-Rad, Hercules, CA, USA). PCR-amplified products were separated using 2% agarose gel, including ethidium bromide (Sigma, St. Louis, MO, USA), and imaged in a gel documentation system (Fujifilm, Tokyo, Japan) [30]. The following primer sequences were used: mouse tyrosinase 5′-ATAACAGCTCCCACCAGTGC-3′ (sense) and 5′-CCCAGAAGCCAATGCACCTA-3′ (antisense) (NM_011661.5); mouse MITF, 5′-CTGTACTCTGAGCAGCAGGTG-3′ (sense) and 5′-CCCGTCTCTGGAAACTTGATCG-3′ (antisense) (NM_001178049.1); mouse TRP-1 5′-AGACGCTGCACTGCTGGTC AAGCCTGTAGCCCACGTCGTA-3′ (sense) and 5′-GCTGCAGGAGCCTTCTTTCT-3′ (antisense) (NM_001282015.1). The expression of glyceraldehyde 3-phosphate dehydrogenase was used as an endogenous control for qRT-PCR experiments [20].

### 2.16. Statistical Analysis

IBM SPSS online version 26.0 (SPSS, Inc., Chicago, IL, USA, an IBM company) was used for statistical analysis. A one-way analysis of variance was used to assess the statistical significance of the differences among treatment groups. For each statistically significant effect of treatment, Duncan’s multiple range test was used for comparisons between multiple group means. The data were expressed as mean ± standard deviation (SD).

## 3. Results and Discussion

### 3.1. Isolation and Yield Results of CCSB

The suspension of 269.03 g of 70% EtOH extract of CCSB in distilled water yielded n-hexane (n-Hex), chloroform (Ch), ethyl acetate (EtOAc), n-butanol (*n*-BuOH), and water (H_2_O) fractions of 4.84%, 1.96%, 19.04%, 16.63%, and 5.92%, respectively, indicating that the EtOAc fraction had the highest yield (Figure 1). These findings also indicate that polar solvent molecules predominate in CCSB extract. A white powder containing an ester carbonyl group and a phenolic molecule was extracted as Compound 1. Its ^13^C-NMR spectrum (DMSO-d_6_, 100 MHz) analysis revealed nine carbon signals, containing a total of seven signals. The identification of an ester group at δC 166.31 (C-7), δC 108.92 (C-2,6), δC 120.03 (C-1), δC 138.82 (C-4), and δC 146.02 (C-3,5) confirmed the presence of an aromatic ring. A 3′ alkyl group was identified at δC 14.72 (C-9), and an ether group was identified at δC 60.48 (C-8). One symmetry peak was detected at δH 6.88 (2H, singlet, H-2,6) of ^1^H-NMR (DMSO-d_6_, 400 MHz), and the ethyl groups at δH 4.16 (2H, quartet, H-8) and δH 1.22 (3H, triplet, H-9) suggested the presence of a 3,4,5-trisubstituted pattern (Appendix A). Consequently, the structure of Compound 1 was verified as EG (Figure 2) based on a comparison with the relevant literature [31]. Notably, EG is a food additive with the E number E313 and is the ethyl ester of gallic acid. It is usually added to food as an antioxidant. As a potential antioxidant compound, EG is currently attracting considerable research interest. EG is also one of several phenolic compounds, including catechin, quercetin, kaempferol, apigenin, and naringenin [32,33].

### 3.2. Antioxidant Activity Results of CCSB and Other Fractions

#### 3.2.1. DPPH Radical Scavenging Activity

The antioxidant activity of natural products is commonly measured by their ability to neutralize the DPPH free radical, a rather stable free radical that causes cell aging and many diseases [34]. In this experiment, the DPPH free radical scavenging ability was measured for the 70% EtOH extract of CCSB and each solvent fraction. Without using standards, sample concentrations were tested from 25 to 1000 g/mL, and their respective IC_50_ values were determined (Figure 3A, Table 1). The DPPH free radical scavenging ability of CCSB extracts and fractions was found to increase concentration-dependently, and the antioxidant activity of each fraction was in the order of ethyl acetate, chloroform, n-butanol, water, and n-hexane. The IC_50_ value of the EtOAc fraction was 77.77 ± 1.79 μg/mL, which demonstrated excellent DPPH radical scavenging activity, corroborating the result of an experiment in which the EtOAc fraction exhibited the highest antioxidant activity when 80% ethanol extract of CCSB was fractionated into n-hexane, dichloromethane, ethyl acetate, *n*-butanol, and water by Kim et al. [35]. In the study reported by Kalaivani et al., the DPPH radical scavenging activity was measured by isolating EG from leaves of *Acacia Nilotica* (L.) Wild. Ex. Delile Subsp. *Indica* (Benth.) Brenan [36]. This measurement revealed that, as the EG concentration increased, the DPPH radical scavenging activity also increased. In addition, EG had a higher DPPH radical scavenging activity than ascorbic acid, which was used as the positive control.

#### 3.2.2. ABTS Radical Scavenging Activity

The chemical reaction of ABTS results in a transition from blue-green to transparent when radical cations gain electrons from antioxidants [37]. In this experiment, the ABTS cation radical scavenging activity was measured for the 70% EtOH extract of CCSB and each solvent fraction. Without using standards, sample concentrations were tested from 25 to 1000 g/mL, and their respective IC_50_ values were determined (Figure 3B, Table 1). Notably, the ABTS cation radical scavenging ability of CCSB extracts and fractions increased concentration-dependently, and the antioxidant activity of each fraction was in the order of ethyl acetate, chloroform, n-butanol, water, and *n*-hexane. The IC_50_ value of the ethyl acetate fraction was 57.43 ± 0.40 μg/mL, showing excellent ABTS radical scavenging activity. The experimental results indicated that the ABTS radical had a significantly higher IC_50_ value than the DPPH radical, based on the ability to measure the antioxidant capacity of hydrophilic substances and lipophilic compounds by dissolving them in water and organic solvents [38].

### 3.3. Total Polyphenol and Total Flavonoid Contents

Owing to their phenolic structures and double bonds, polyphenols, which are abundant in plant resources, have antioxidant, anti-inflammatory, antibacterial, and anticancer activities [39]. Plants generate polyphenols in minute quantities as secondary metabolites, and the structure of polyphenol compounds is stable even when dehydrogenation occurs due to the availability of electrons, which induces reduced characteristics and functions as antioxidants [40]. In this experiment, a standard calibration curve was prepared using gallic acid as the standard material, and the total polyphenol content of the 70% EtOH extract of CCSB and each solvent fraction was measured. The total polyphenol content of the extracts and fractions was expressed in terms of gallic acid equivalent (GAE) per mg/g of weight (Table 1). The highest polyphenol content was found in the EtOAc fraction (368.87 ± 3.22 GAE mg/g). Therefore, CCSB contains more polyphenols than its fruits, considering Lee et al.’s study in which total phenolic content obtained from CCS fruits by extraction method using tannic acid as a standard substance reached a maximum of 27.69 mg% in water extract [41].

Furthermore, a standard calibration curve was prepared using quercetin as a standard material, and the total flavonoid content contained in the 70% EtOH extract of CCSB and each solvent fraction was measured. The total flavonoid content of the extracts and fractions was expressed in terms of quercetin equivalent (QUE) per 1 g of weight (Table 1). The highest flavonoid content was found in the EtOAc fraction (45.14 ± 0.73 QUE mg/g).

### 3.4. Polyphenol Content Analysis Using HPLC MS/MS

Here, 16 types of polyphenols were identified via HPLC MS/MS, and the polyphenol content of the EtOAc fraction, which had the highest antioxidant activity among the fractions isolated from the 70% EtOH extract of CCSB, was analyzed. The polyphenols identified through HPLC MS/MS analysis were 4-hydroxybenzoic acid, caffeic acid, syringic acid, vanillic acid, coumaric acid, ferulic acid, naringenin, benzoic acid, nicotinic acid, gallic acid, protocatechuic acid, chlorogenic acid, catechin, kaempferol, EG, and epigallocatechin gallate (Figure 4). Among them, EG had the highest content at 330.00 mg/g, followed by gallic acid and chlorogenic acid. Notably, ellagic acid and its derivatives, as well as hexahydroxydiphenoyl esters and phenazine derivatives, were identified in CCS leaves [42,43]. Gallic acid, a well-known secondary metabolite of biosynthesis, was also found in high concentrations of ellagic acid and its derivatives in CCS leaves [44]. Consequently, ellagic acid and its derivatives are also present in CCSB, giving it a relatively high gallic acid content. A polyphenol analysis of CCSB leaf extract revealed that it contained large amounts of epigallocatechin gallate, EG, ρ-coumaric acid, and caffeic acid, corroborating the findings of the present study [45].

### 3.5. Results of EG Tyrosinase Inhibitory Activity Using a Cell-Free System

Tyrosinase, which catalyzes the rate-determining step of melanin biosynthesis, is an enzyme used to evaluate the whitening activity effect. To investigate the effect of CCSB on tyrosinase activity inhibition, the tyrosinase inhibitory activity of the extract fractions (CCSB-Hex, CCSB-Ch, CCSB-EtOAc, CCSB-BuOH, CCSB-Water) was measured. Among these fractions, CCSB-EtOAc had the highest inhibitory activity, with an IC50 value of 217.29 ± 1.76 µg/mL (Figure 5). Ascorbic acid and EG were used as positive controls, and their IC_50_ values were 30.99 ± 0.86 and 71.27 ± 0.38 µg/mL, respectively.

### 3.6. Enzyme Kinetics Analysis of Tyrosinase

The reaction rate of the entire reaction system usually determines the Km value of an enzymatic reaction. In this study, we calculated Km and Vmax and identified the type of inhibition using the Lineweaver–Burk equation. The inhibition pattern of an enzyme depends on the binding site and type of binding mode. During competitive inhibition, the inhibitor competitively binds to the substrate noncovalently, thereby inhibiting enzyme activity. In noncompetitive inhibition, the inhibitor reversibly binds to both the free enzyme and enzyme–substrate complex to exhibit inhibitory effects. In this study, the values of Km, Vmax, and inhibition constant (Ki) of EG against tyrosinase were 1.9675 mM, 0.1175 mmol/min, and 0.4341 mM, respectively (Figure 6). The Lineweaver–Burk plot was linear, confirming that the kinetic behavior was noncompetitive.

### 3.7. Molecular Docking Study

Protein–ligand docking attempts to predict the location and orientation of the ligand when it binds to a protein receptor or enzyme [24]. We used molecular docking to predict the binding between EG and human tyrosinase. Met 374, Ser 380, His 180, His 211, His 390, His 363, Phe 347, Asn 364, and Ile 368 surrounded hydrophobic pockets, as predicted by the binding model (Figure 7A). In these hydrophobic pockets, EG formed ligand conformations that inhibited enzyme activity. When a ligand was formed between tyrosinase and EG, hydrogen bond interactions were formed at Met 374, Ser 380, and His 390. The docking results revealed that EG can bind to the active site of tyrosinase. Based on the ligand interaction diagram of EG, hydrogen bond interactions were involved with the Met 374 and Ser 380 residues of tyrosinase, and a pi–pi interaction was possible between EG and His 367 of tyrosinase (Figure 7B) [46].

### 3.8. Cytotoxicity Evaluation and Quantitative Analysis of Intracellular ROS

Mitochondrial dehydrogenase and MTT tetrazolium react during cell metabolism to form MTT formazan, which turns purple. Using this principle, cell viability can be measured [47]. For each concentration (1–100 µg/mL), EG cytotoxicity was determined to be <5 µg/mL (Figure 8A). Jin et al. (2006) isolated EG from *Acer okamotoanum* Nakai and measured its cytotoxicity against B16F10 cells [48]. The cytotoxicity IC_50_ value of EG isolated from *A. okamotoanum* was approximately 9.29 μg/mL. In this study, cell viability was significantly reduced compared to the control at concentrations of at least 10 μg/mL. Changes in the intracellular ROS content of B16F10 after EG treatment were measured. When EG was treated at concentrations of 1, 2.5, and 5 μg/mL, the intracellular ROS significantly decreased compared to the group treated with only t-BHP (Figure 8B).

### 3.9. Results of Melanin Content by EG in B16F10 Cells

Mushroom-derived tyrosinase is 23% identical in amino acid sequence to human tyrosinase, and mouse-derived tyrosinase is 82% similar in sequence identity to human tyrosinase, so the experiment is usually more accurate [49]. When B16F10 cells were treated with α-MSH (1 μg/mL) and EG, the melanin content reduced proportionally to the EG concentration (Figure 9). The melanin production inhibitory activity was similar to that of 2.5 μg/mL EG and arbutin, which were positive controls. Regarding 5 μg/mL EG treatment, EG inhibitory activity surpassed that of arbutin.

### 3.10. Effect of EG on Antimelanogenesis-Related Proteins/Genes and Autophagy-Related Proteins in the B16F10 Cell Line

Melanin is synthesized via various intracellular signaling pathways, including the cAMP/PKA pathway. UV boosts cAMP levels in melanin cells, activates PKA, and promotes MITF expression via CREB. MITF is an important transcriptional regulator in the melanin synthesis pathway, promoting the transcription of TYR, TRP-1, and TRP-2. EG was isolated from CCSB, and its whitening activity was confirmed by measuring the expression levels of TYR, TRP-1, TRP-2, and MITF proteins in B16F10 cells stimulated by α-MSH. When B16F10 melanoma cells were treated with EG at 1, 2.5, and 5 μg/mL, the expression levels of TYR, TRP-1, TRP-2, MITF, ρ-PKA, PKA, ρ-CREB, and CREB proteins decreased EG concentration-dependently (Figure 10). In particular, at 5 μg/mL, an antimelanogenic effect similar to that of arbutin was confirmed. Kim et al. (2018) extracted 75% EtOH of CCS to measure tyrosinase inhibition in B16F10 cells [50]. They found that the extract exhibited 37.9% tyrosinase inhibition activity, exceeding that of the positive control arbutin (33.9%). Many studies have also explored the inhibition of melanin production using natural resources. Li et al. (2018) extracted *Morus alba* L. leaves with 70% EtOH; after fractionalization, the extract was used to treat B16F10 cells stimulated by α-MSH [51]. They found that the studied extract dose-dependently lowered the expression levels of CREB, MITF, TYR, and TRP-1 in B16F10 cells. Pedrosa et al. (2016) also reported the tyrosinase inhibition activity of 50% EtOH extract of *Libidibia ferrea* Mart in B16F10 cells [52]. In an effort to identify novel cosmetic materials with antiaging and whitening properties, research on the production of materials that control the autophagy process has been intensively conducted in recent years. Autophagy is a revolutionary antiaging concept that is necessary for cells to eliminate intracellular wastes and old proteins, in addition to recycling and rebuilding them into healthy cells. Importantly, inducing autophagy has been shown to suppress melanin production, suggesting a connection between these two processes. ATP is required for AMP-activated protein kinase (AMPK) activation and inactivation of mTOR’s target [53]. In addition, LC3B-II, an important protein marker in autophagy, induces ERK activation; ERK increases MITF expression via CREB phosphorylation; and ATG7 and Beclin-1 are positively related to MITF expression and MITF transcriptional activity, respectively [54]. To investigate EG autophagy-enhancing activity, AKT, mTOR, Beclin, and LC3B autophagy markers were induced by treating B16F10 cells with MSH, and the protein expression patterns were then measured. As a result, the protein expression of AMPK and p-AKT of the protein markers related to autophagy was increased, and the protein expression of p-m-TOR was decreased (Figure 11). In addition, Beclin and LC3B expression levels both increased, confirming that autophagy was activated. *Patrinia villosa* (Thunb.) Juss extract prevented melanogenesis and induced autophagy via autophagy markers LC3B and p62 [10]. A study described a mechanism underpinning the regulation of melanin synthesis or function in antiaging, finding that autophagy responded in accordance with AKT2 concentrations [54].

Based on Western blotting data, the expression levels of melanin production-related genes, TYR, TRP-1, TRP-2, and MITF, were evaluated at the mRNA level using RT-PCR in order to confirm EG skin-whitening activity. α-MSH induced TYR, TRP-1, TRP-2, and MITF expression mediated by EG, whereas EG treatment reduced mRNA expression (1–5 µg/mL) concentration-dependently. In particular, 5 µg/mL of EG inhibited mRNA expression more effectively than the positive control arbutin (Figure 12).

We have demonstrated that autophagy enhancers can contribute to inhibiting the melanogenic pathway (Figure 13), but autophagy inhibition or blockage may be required for antimelanogenesis to be effective. To determine the effect of autophagy flux on melanogenesis signaling, further research is required.

## 4. Conclusions

This study demonstrated that EG isolated from *Castanopsis cuspidata* var. *sieboldii* branch (CCSB) extract promoted autophagy and inhibited melanogenesis. AMPK and p-AKT increased in the α-MSH-related pathway, with an enhanced LC3B signal in autophagy flux. EG significantly reduced the levels of whitening-associated proteins, p-CREB, and p-PKA, and it inhibited the expression of TYR, TRP-1, TRP-2, and transcription factor (MITF) in a dose-dependent manner. Therefore, CCSB extract could serve as a novel functional cosmetic material with antimelanogenic and autophagy-enhancing activity. We plan to obtain soluble expression of human tyrosinase and semisynthesis of novel derivatives of EG. By combining the result of this and future studies, EG and EG derivatives can be established as novel cosmetic ingredients with autophagy-regulating properties once the clinical study is completed.

## Figures and Tables

**Figure 1 antioxidants-12-00269-f001:**
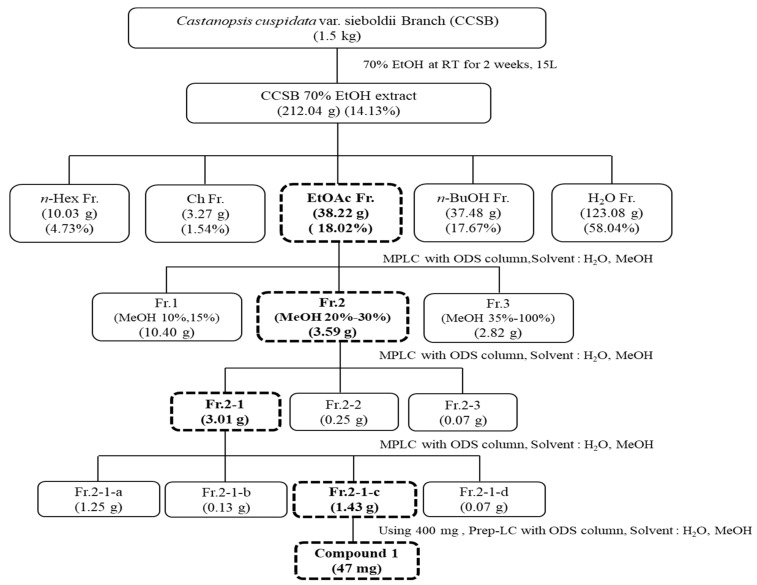
Isolation diagram of *C. cuspidata* var. *sieboldii* branch EtOAc fraction and subfraction. n-Hex fr.: n-hexane fraction, Ch Fr.: chloroform, EtOAc Fr.: ethyl acetate fraction, n-BuOH Fr.: n-butanol fraction, and H_2_O Fr.: water fraction, Fr.1: ethyl acetate fraction 1, Fr.2: ethyl acetate fraction 2, Fr.3: ethyl acetate fraction 3, Fr.2-1: ethyl acetate fraction 2-1, Fr.2-2: ethyl acetate fraction 2-2, Fr.2-3: ethyl acetate fraction 2-3, Fr.2-1-a: ethyl acetate fraction 2-1-a, Fr.2-1-b: ethyl acetate fraction 2-1-b, Fr.2-1-c: ethyl acetate fraction 2-1-c, Fr.2-1-d: ethyl acetate fraction 2-1-d.

**Figure 2 antioxidants-12-00269-f002:**
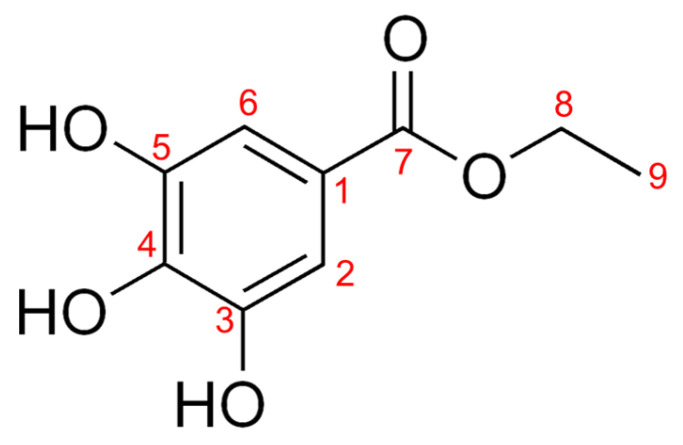
Chemical structure of Compound 1 (ethyl gallate).

**Figure 3 antioxidants-12-00269-f003:**
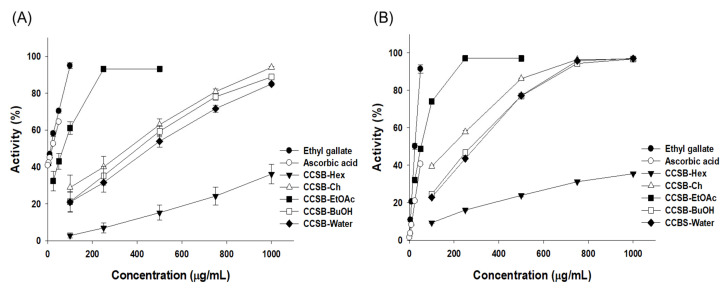
(**A**) DPPH free radical scavenging activities and (**B**) ABTS cation radical scavenging activities of extract and solvent fractions from *C. cuspidata* var. *sieboldii* branch (CCSB).

**Figure 4 antioxidants-12-00269-f004:**
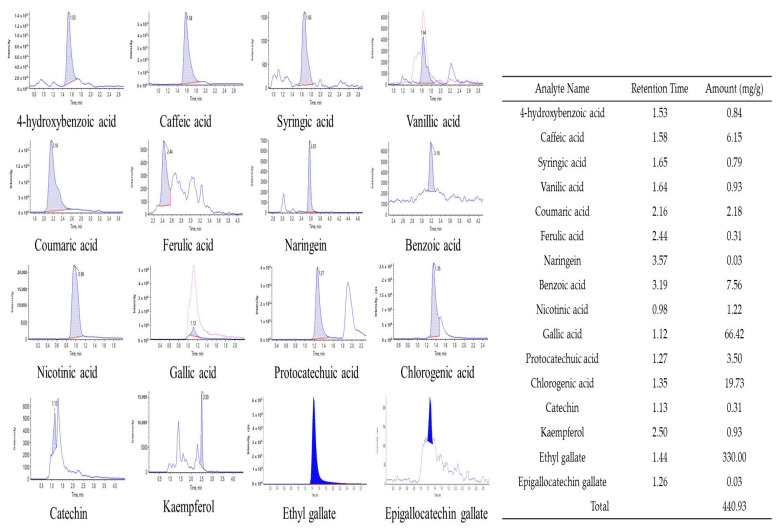
Component analysis of *C. cuspidata* var. *sieboldii* branch EtOAc fraction by HPLC MS/MS.

**Figure 5 antioxidants-12-00269-f005:**
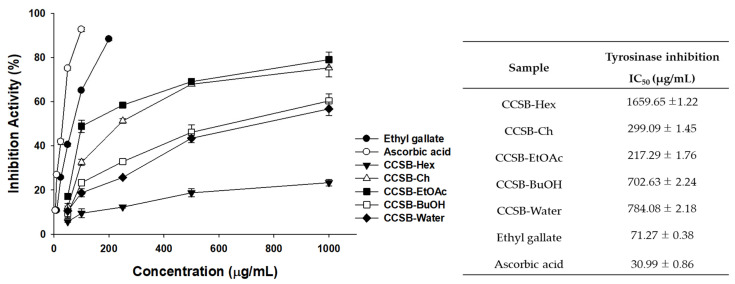
The IC_50_ value of tyrosinase inhibition activity of the extract and solvent fractions from *C. cuspidata* var. *sieboldii* branch (CCSB). Each experiment was assayed in triplicate. Data were represented as means ± SD.

**Figure 6 antioxidants-12-00269-f006:**
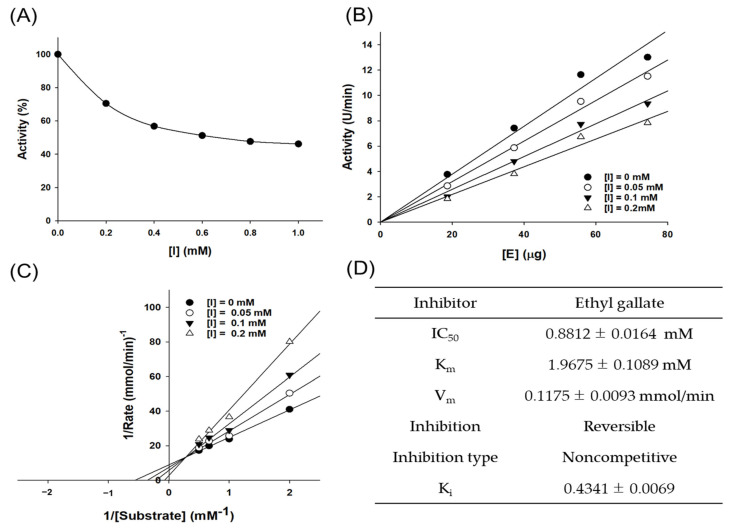
(**A**) Effect of EG on tyrosinase activity. (**B**) Tyrosinase–EG concentration relationship in catalytic activity. EG concentrations for curves 1–4 were 0, 0.05, 0.1, and 0.2 mM, respectively. (**C**) Lineweaver–Burk plots for EG inhibition on tyrosinase. EG concentrations for curves 1–4 were 0, 0.05, 0.1, and 0.2 mM, respectively. (**D**) Kinetic parameters and microscopic inhibition rate constants of mushroom tyrosinase in the presence of EG.

**Figure 7 antioxidants-12-00269-f007:**
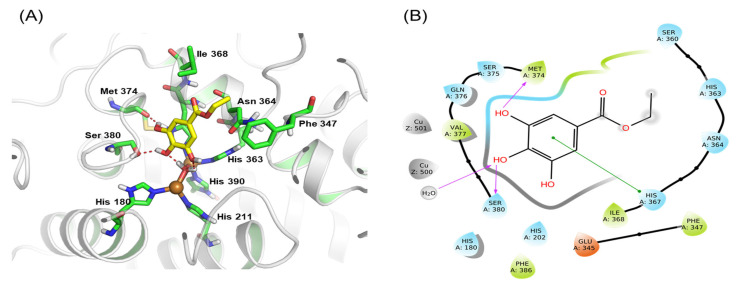
(**A**) Binding model prediction of human tyrosinase and ethyl gallate. The red dashed line denotes the hydrogen bond interaction. (**B**) Ligand interaction diagram of human tyrosinase and ethyl gallate. The purple arrows denote hydrogen bond interactions, and the green color denotes pi–pi interactions of human tyrosinase and ethyl gallate.

**Figure 8 antioxidants-12-00269-f008:**
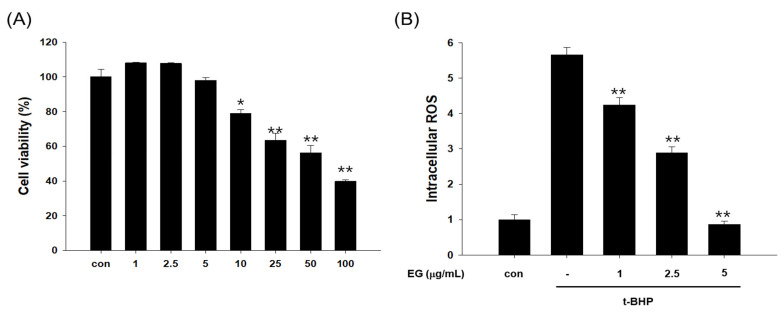
Measurement of (**A**) viability of B16F10 melanoma cells treated with ethyl gallate and (**B**) intracellular ROS production. Ethyl gallate: 1–5 μg/mL (5.05–25.23 μM); * *p* < 0.05, ** *p* < 0.01, compared with *t*-BHP treatment.

**Figure 9 antioxidants-12-00269-f009:**
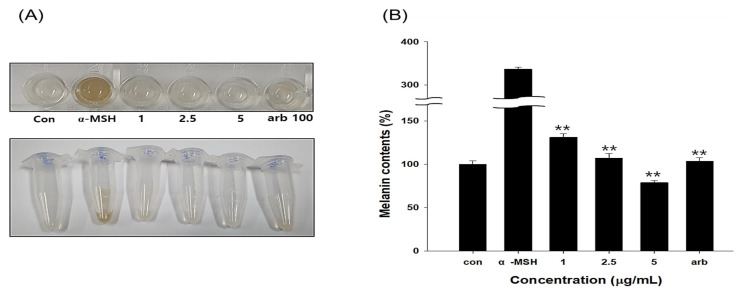
Measurement of melanin content in B16F10 melanoma cells treated with ethyl gallate. (**A**) B16F10 melanoma cells treated with ethyl gallate. The cells were cultured for 48 h in the presence of 1–5 μg/mL (5.05–25.23 μM) ethyl gallate as a positive control or 1 μg/mL α-MSH. (**B**) Measurement of melanin content with 1–5 μg/mL ethyl gallate (5.05–25.23 μM). Relative melanin content was determined 72 h after treatment. *n* = 3, error bars, mean ± standard deviation. Effect of 100 μg/mL arbutin (367 μM) on melanin synthesis and tyrosinase activity in B16F10 cells. Significantly different compared with α-MSH, ** *p* < 0.01. α-MSH: α-melanocyte-stimulating hormone.

**Figure 10 antioxidants-12-00269-f010:**
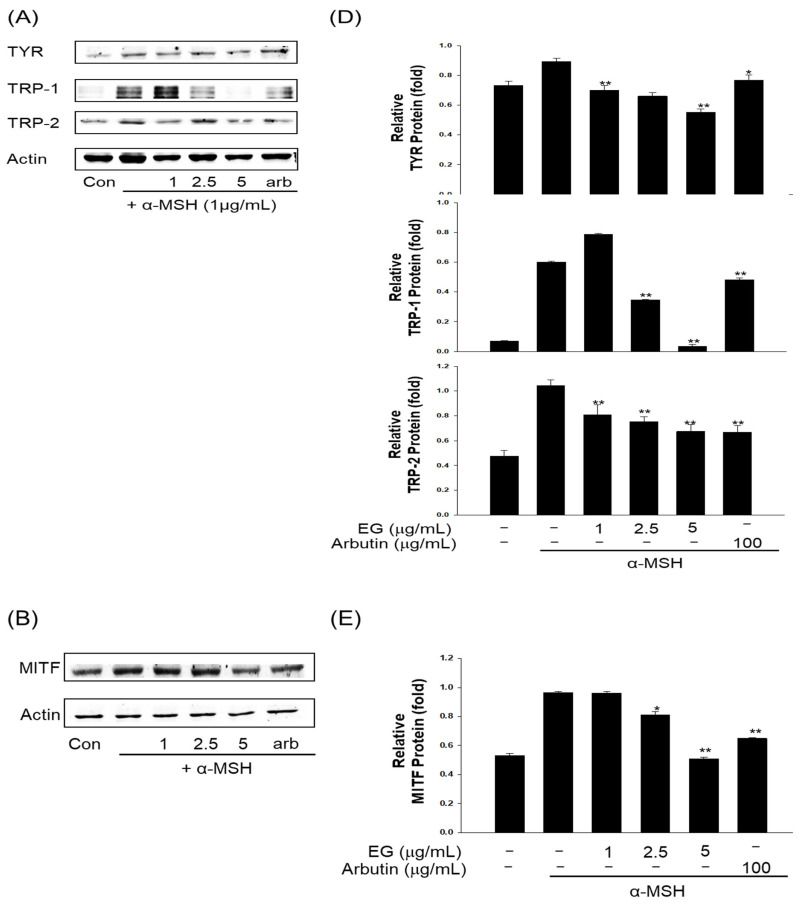
(**A**) Effect of ethyl gallate (EG) and arbutin on tyrosinase (TYR), tyrosinase-related protein 1 (TRP-1), and tyrosinase-related protein 2 (TRP-2) protein expression in B16F10 melanoma cells. The cells were treated with the indicated concentrations of EG and arbutin prior to α-melanocyte-stimulating hormone (α-MSH) treatment for 24 h. The loading control was assessed using a β-actin antibody. (**B**) Effect of EG and arbutin on microphthalmia-associated transcription factor (MITF) protein expression in B16F10 melanoma cells. The cells were treated with the indicated concentrations of EG and arbutin prior to α-MSH treatment for 4 h. (**C**) Effects of EG and arbutin on phosphorylated protein kinase A (p-PKA), PKA, phosphorylated cAMP response element binding (p-CREB), and CREB protein expression in B16F10 melanoma cells. The cells were treated with the indicated concentrations of EG and arbutin prior to α-MSH treatment for 3 h. (**D**) Quantitative analysis of TYR, TRP-1, and TRP-2 by Western blotting. (**E**) Quantitative analysis of MITF by Western blotting. (**F**) Quantitative analysis of p-PKA, PKA, p-CREB, and CREB by Western blotting. * *p* < 0.05, ** *p* < 0.01, compared with α-MSH treatment.

**Figure 11 antioxidants-12-00269-f011:**
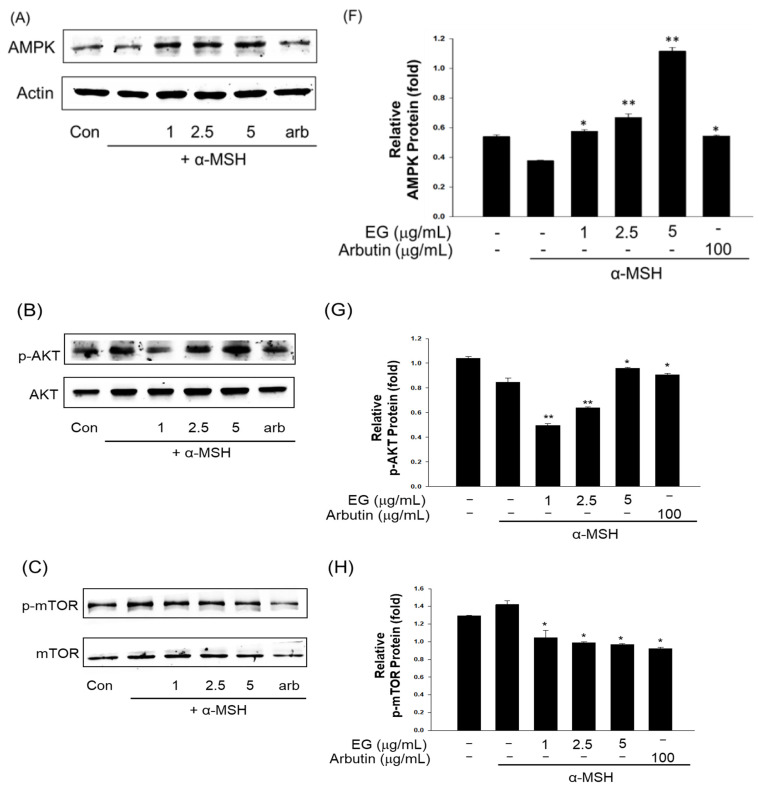
(**A**) Effect of EG and arbutin on AMP-activated protein kinase (AMPK) protein expression in B16F10 melanoma cells. The cells were treated with the indicated concentrations of EG and arbutin prior to α-MSH treatment for 4 h. (**B**) Effect of EG and arbutin on phosphorylated AKT (p-AKT) protein expression in B16F10 melanoma cells. The cells were treated with the indicated concentrations of EG and arbutin prior to α-MSH treatment for 4 h. (**C**) Effect of EG and arbutin on the phosphorylated mammalian target of rapamycin (p-mTOR) and mTOR protein expression in B16F10 melanoma cells. The cells were treated with the indicated concentrations of EG and arbutin prior to α-MSH treatment for 4 h. (**D**) Effect of EG and arbutin on Beclin protein expression in B16F10 melanoma cells. The cells were treated with the indicated concentrations of EG and arbutin prior to α-MSH treatment for 4 h. (**E**) Effect of EG and arbutin on LC3B protein expression in B16F10 melanoma cells. The cells were treated with the indicated concentrations of EG and arbutin prior to α-MSH treatment for 4 h. (**F**) Quantitative analysis of AMPK by Western blotting. (**G**) Quantitative analysis of p-AKT by Western blotting. (**H**) Quantitative analysis of p-mTOR and mTOR by Western blotting. (**I**) Quantitative analysis of Beclin by Western blotting. (J) Quantitative analysis of LC3B by Western blotting. * *p* < 0.05, ** *p* < 0.01, compared with α-MSH treatment.

**Figure 12 antioxidants-12-00269-f012:**
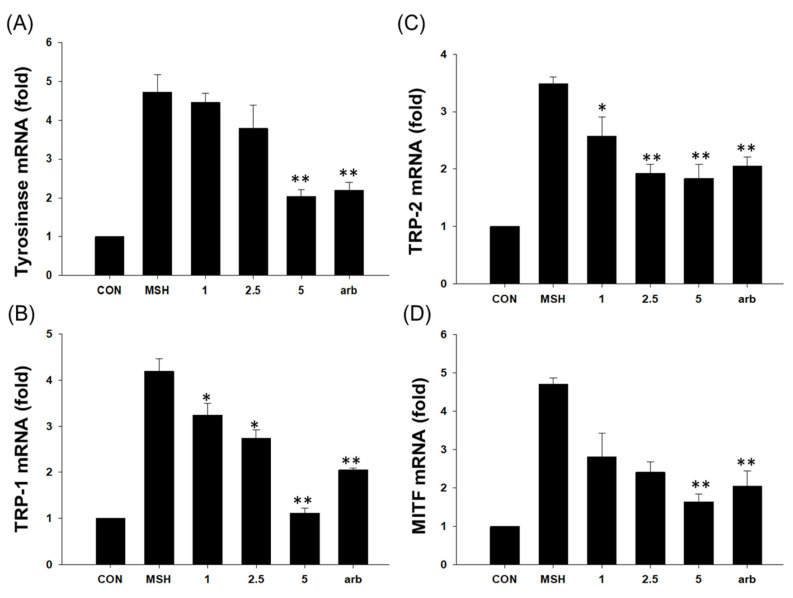
RT-PCR analysis of EG. (**A**) Effects of EG on tyrosinase, (**B**) TRP-1, (**C**) TRP-2, and (**D**) MITF mRNA in B16F10 cells. Data from separate experiments were presented (statistically significant group vs. the vehicle-treated control, * *p* < 0.05, ** *p* < 0.01, bars indicate SD). α-MSH: α-melanocyte-stimulating hormone, MITF: microphthalmia-associated transcription factor, TYR: tyrosinase.

**Figure 13 antioxidants-12-00269-f013:**
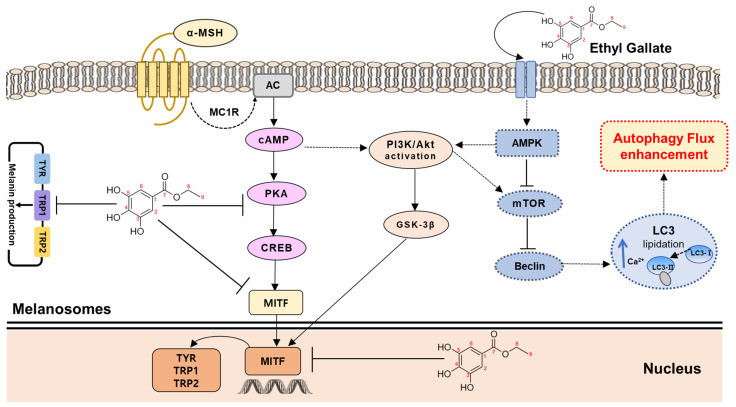
Schematic diagram of the signaling pathway of antimelanogenesis–autophagy correlation.

**Table 1 antioxidants-12-00269-t001:** Antioxidant effect and total polyphenol and total flavonoid content results of CCSB.

Sample	DPPH IC_50_ (μg/mL)	ABTS IC_50_ (μg/mL)	TPC (GAE mg/g)	TFC (QUE mg/g)
CCSB-Hex	2038.59 ± 43.88	1443.51 ± 12.74	189.45 ± 0.14	21.78 ± 2.55
CCSB-Ch	366.90 ± 4.28	176.35 ± 5.09	181.66 ± 6.51	33.59 ± 1.67
CCSB-EtOAc	77.77 ± 1.79	57.43 ± 0.40	368.87 ± 3.22	45.14 ± 0.73
CCSB-BuOH	434.72 ± 1.24	300.63 ± 1.36	123.22 ± 0.72	13.00 ± 0.36
CCSB-Water	483.58 ± 2.17	313.20 ± 1.44	102.92 ± 0.14	13.63 ± 0.36
Ethyl gallate	15.45 ± 0.26	25.55 ± 0.78	-	-
Ascorbic acid	20.26 ± 0.33	59.63 ± 0.19	-	-

TPC: total polyphenol contents, GAE: gallic acid equivalent, TFC: total flavonoid content, QUE: quercetin equivalent.

## Data Availability

Data is contained within the article and Appendix A.

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
