# Peer review of "Ethyl Gallate Isolated from Castanopsis cuspidata var. sieboldii Branches Inhibits Melanogenesis and Promotes Autophagy in B16F10 Cells"

_antioxidants, 2023, doi:10.3390/antiox12020269_

Round 1

Reviewer 1 Report

The paper is dedicated to the fractionation of Castanopsis cuspidata extracts and the evaluation of different biological activities.

The paper is interesting, however, some improvements are necessary. My suggestions:

- add purities and/or manufacturer of all chemicals and other materials used in this study.

- the term "sieboldii" must be in italic.

- explain abbreviations used in Fig. 1 in the caption of this figure.

- indicate the temperature used for the drying of plant material (line 62).

- provide an amount of water and other solvents used for the fractionation.

- line 87: are you sure about 570 min?

- clarify the procedure and sample preparation of NMR analysis.

- which solvent was used for DPPH preparation (106 line)?

- provide the range of standard concentrations used for calibration curves (gallic acid, quercetin, ...). 

- add data of un-fractionated extract to the Tables and Figures were fractions are presented.

- how was quantitative analysis performed of phenolic compounds?

- Fig. 6 - standard deviations must be added

- it was not clear "-" in Fig. 10 D, E, F and Fig. 11

- add more details in the conclusions.

Author Response

Reply to Reviewers

Thank you very much for your comments for the manuscript. Here are the answers to reviewer’s comments. Please find the file attached.

Regards,

Hyun-Jae Shin

Reviewer 2 Report

Recently, much attention has been paid to the identification of antioxidant and anti-inflammatory properties of natural plant extracts. The article of Choi et al also belongs to such type of works. A set of advanced methods, including different types of chromatographic analysis, NMR of the active components, RT-PCR allows the authors to obtain a number of different extracts from branches of Castanopsis cuspidate, analyze its polyphenol and flavonoid content and to isolate an effective poliphenol antioxidant ethyl gallate (EG). Among antioxidant properties EG proved to be an antimelanogenic factor inhibiting tyrosinase, the enzyme which controls rate-limiting step of melanin synthesis. The article makes a good impression. Radical scavenging and tyrosinase inhibition have been reliably shown for both EG and individual extracts. Inhibition of melanin production was also shown on Bl6F10 melanoma cell culture stimulated by α-MSH.

At the same time, there are a number of shortcomings that need to be corrected. Figure 5 should demonstrate the inhibition of tyrosinase by EG and extracts, but for some reason the ordinate axis is marked as “activity”. It should be explained what is meant by the term activity in this case. Considering antimelanogenic effects of EG and extracts the authors suppose to use it as cosmetic material. To this aim evaluation of viability effect of EG and extracts on normal cells (fibroblast culture) is very important and was not done. It was found that EG is viable within 1-5µg/ml for Bl6F10 melanoma cells (Fig.8). It is rather toxic and establishing a safe dose of EG use on ordinary cells is becoming even more urgent. On the other hand the toxicity of EG for melanoma cells might be used to destroy this harmful cancer and such possibility deserves discussion. On the same Fig.8 the effect of EG is compared to t-BHP, the abbreviation which should be decoded. Fig.9 represents melanin content in Bl6F10 cells treated with different doses of EG compared to arbutin (100µg/ml) but it is quite difficult to notice dose dependence because the differences between columns are leveled by huge effect of  α-MSH treatment. To elucidate dose effect I should recommend to make a break in the ordinate axis from 300 to 270. Besides it is necessary to validate the dose of arbutin used in the text section and to include the corresponding reference. I recommend the article for publication after all corrections mentioned above.

Author Response

(The authors gave the same response as above.)

Reviewer 3 Report

1) The authors in the introduction suggest that they have isolated ethyl gallate ("Notably, prior to this study, neither the isolation of ethyl gallate (EG) nor elucidation of whitening mechanisms related to autophagy signals had been described"), while this information should only found in the results section.

2) The authors suggest that the antioxidant activity of ethyl gallate is poorly understood. In fact, this relationship has already been studied: DOI: 10.1111/j.1750-3841.2011.02243.x this work should be included in the discussion.

3) I think the authors can also mention their previous research: https://doi.org/10.3390/ijms19020409

4) I have doubts about figure 4. Does it show chromatograms of individual compounds? Why didn't the Authors show the whole chromatogram? Was the separation of the ingredients sufficient? For example, caffeic, syringic and vanillic acids have practically the same retention times...

Author Response

(The authors gave the same response as above.)

Round 2

Reviewer 3 Report

The authors clarified my doubts and revised the manuscript as recommended. I consider it suitable for publication in Antioxidants.